# DeepQuali: Initial results of a study on the use of large language models for assessing the quality of user stories

**Abstract.**

Generative AI, especially large language models, is increasingly used in software engineering, but requirements validation remains underexplored. Existing work focuses on eliciting, transforming, and classifying requirements rather than assessing their quality. We present DeepQuali, an LLM-based approach for assessing and improving requirements quality in agile projects. Applied in two small companies, we compared DeepQuali's evaluations with expert judgments and collected feedback in walkthroughs. Experts largely agreed with DeepQuali's overall ratings and explanations, but not always with each other on detailed assessments. They found the approach useful yet insufficiently integrated into their workflow. Explicit quality models and explanatory feedback increased acceptance.

**Keywords:** agile software engineering, requirements quality assurance, large language models.

## 1 Introduction

Software engineering (SE) practitioners increasingly recognize Generative AI's (GAI) potential to automate traditionally labor-intensive and error-prone development tasks [18]. Automation in SE was considered challenging due to the creative and unstructured nature of artifacts such as requirements, design models, code, and documentation. GAI's ability to process unstructured data and interpret context has created new opportunities, quickly adopted by SE research and practice [18].

The use of GAI, especially Large Language Models (LLMs, is rapidly expanding across all phases of the software development lifecycle (SDLC). Most SE use cases focus on coding tasks like code generation and defect detection. In contrast,

requirements engineering (RE) has received less attention despite its unstructured artifacts and lack of labeled data, which make it ideal for GAI application.

LLM use cases in RE [23] currently address constructive stages like elicitation, analysis, and specification [27]. Requirements validation, which assesses the quality of requirements specifications, remains the least automated RE stage. Automated validation is especially important in agile software development (ASD), where user stories change frequently and cycles are short.

**Contribution**: We evaluate DeepQuali, an LLM-based approach for assessing user story quality using explicit quality models. DeepQuali provides numeric quality scores, descriptions of quality issues, explanations, and improvement suggestions. The evaluation in two small software development companies focuses on user feedback regarding quality assessments and overall usefulness in ASD projects.

The remainder of this paper is organized as follows: Section 2 reviews background and related work on requirements quality assurance (QA), including GAI applications. Section 3 introduces DeepQuali for assessing user story quality. Section 4 outlines the evaluation methodology. Section 5 summarizes and discusses the evaluation results. Section 6 addresses threats to validity. Section 7 concludes the paper.

## 2    Background and Related Work

High-quality requirements are essential for successful software projects, improving stakeholder communication, supporting testing, and enabling reliable planning. Poor requirements lead to costly rework, especially when detected late. Various methods ensure quality: constructive approaches (e.g., structured interviews, focus groups [3], prototyping [28], formal specification [4]) build quality in during development, while analytical methods (e.g., inspections and walkthroughs [20], automated checks [6], ambiguity detection, traceability checks [26], formal modeling [4]) evaluate requirements afterward. Despite diverse tool support [23], RE remains complex and time-consuming.

Machine learning and NLP have successfully automated requirements development [15], with LLMs now applied mainly to elicitation, analysis, and specification [27]. Elicitation involves understanding stakeholder needs through dialogues [7] or analyzing user-generated content [21]. LLMs support interpreting [22], classifying [13], prioritizing [21], linking [10] requirements, distinguishing functional/non-functional requirements [30], and identifying quality aspects [1]. For specification, LLMs transform natural language into structured formats [5]. Validation, checking if requirements meet stakeholder needs, remains the least automated phase [23]. Existing solutions detect specific quality deficits [12, 8] like ambiguity [2], often using frameworks such as Quality User Story [17]. Recent LLM-based approaches assess requirements quality through binary classification [8] or interval ratings [14] and suggest improvements [17]. Evaluations compare LLM assessments against expert ground truth using agreement rates [29], interrater reliability [29], Cohen's coefficients [9], and ML metrics [8, 16].

**Further research is needed** on practical usefulness and user acceptance. Current approaches focus on predefined quality aspects (consistency, ambiguity, completeness) but neglect practitioners' needs for custom criteria, problem explanations, and concrete

solutions. Evaluations should move beyond statistical accuracy to include end-user feedback on perceived usefulness and acceptance.

# 3 LLM-Based Assessment of User Story Quality

To address research needs, we propose "DeepQuali", an LLM-based approach for assessing the quality of user stories. We use an existing LLM, i.e., GPT-4o with a custom-defined set of criteria in a structured and explainable manner. Fig. 1 illustrates the basic components of the approach. With

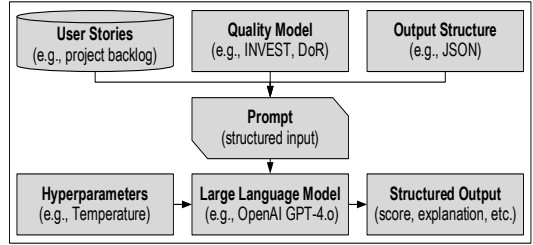

**Fig. 1.** DeepQuali approach

DeepQuali, we envision three major benefits: I) The explicit consideration of a *quality model*, represented, .e.g., by a quality standard (e.g., ISO/IEC 29148 [12]), industrial (e.g., INVEST [26], DoR [20]) or company-specific quality guidelines. II) A comprehensive *output* of the quality assessment, including quality ratings together with their explanation, descriptions of specific quality deficits associated with reference to the quality given guidelines, and corresponding improvement recommendations. III) Structured output assessments according to the predefined *output structure* template.

# 4 Evaluation Methodology

The study's **objective** was to evaluate the practical usefulness and user acceptance of LLM-based quality assessment of user stories in the context of industrial ASD projects from the viewpoint of intended users (i.e., agile software engineers). Specifically, this early evaluation aimed at exploring the practical applicability of DeepQuali, identify its deficits (critical feedback), explain root causes of the critique and identify improvement potentials. From this objective, we derived three **research questions**:

- **RQ1**: To what extent are the quality assessments estimated by DeepQuali consistent with those of human experts?
- **RQ2**: How do intended users evaluate the user story quality assessments provided by DeepQuali?
- **RQ3**: What is the level of acceptance of DeepQuali by intended users?

RQ1 refers to statistical accuracy of the quality assessments, RQ2 to usefulness of the quality assessments from the users' perspectives, and RQ3 to users' overall perceived acceptance of DeepQuali.

## 4.1    Study Context, Constraints, Data and Involved People

DeepQuali and its evaluation emerged from a collaborative research project with two German SMEs specialized in software development. We investigated usage scenarios and stakeholders' needs regarding requirements quality assessment in the **context** ASD. The approach was developed and evaluated using sample user stories with support from software engineering experts from two ASD projects, one developing an online health course portal, the other a vehicle validity data management system. Key **constraints** included data security (sensitive metadata had to be removed from user stories) and limited availability of subject-matter experts. The study design was adapted accordingly, combining on-site workshops with online surveys. Additionally, the immature DeepQuali user interface required guiding users through the application during evaluation. **Data** consisted of sample user stories provided by the companies. **Participants** included four software engineers (two from each company) serving as intended users, with different levels and areas of expertise (cf. Table 1), and the researcher who developed DeepQuali and conducted the evaluation.

**Table 1.** Characteristics of involved experts.

| Characteristic | Company-1 | | Company-2 | |
|---|---|---|---|---|
| | Expert 1 | Expert 2 | Expert-1 | Expert-2 |
| Role | SW engineer | SW Architect | Head of SE | Senior SE |
| Years in current organization | 2 | 4 | 4 | 11 |
| Years at current position | 2 | 13 | 4 | 7 |
| Years in Agile projects | 2 | 7 | 18 | 11 |
| Years specifying user stories | 2 | 7 | 16 | 7 |
| Familiar requirements quality assessment frameworks | None | None | QUS, INVEST | INVEST |
| Familiarity of AI principles | Moderate | Very | Very | Somewhat |
| Practical experience in using GAI | Extensive use | Frequent use | Frequent use | Frequent use |
| GAI methods used | ChatGPT, Github Copilot | ChatGPT, Github Copilot | Chat GPT, self-hosted LLMs | ChatGPT |

## 4.2    Study Procedure and Methods

The evaluation was preceded by two development phases: data acquisition (identifying ASD projects and collecting user stories from backlogs) and development of DeepQuali (optimizing over several train-test iterations). The core evaluation consisted of six steps (cf. Fig. 2):

**(1) Data selection**: We selected five user stories per company with perceived low, medium, and high complexity and quality, then anonymized and transformed them into JSON format.

**(2) Labeling survey**: Experts rated the quality of each story on seven criteria: the six INVEST criteria and a summative Ready-to-Implement (RTI) criterion. Each criterion was operationalized by 3 (INVEST) or 4 (RTI) statements representing sub-aspects from the literature (e.g., [25]). Statements were rated on a 4-point agreement scale (*Strongly disagree – Strongly agree*), mapped to a 1–4 interval without a neutral option to avoid central tendency bias.

**(3) Application of DeepQuali**: We ran DeepQuali using GPT-4o with tuned parameters (e.g., temperature = 0 for stable, consistent outputs). Using the OpenAI API's

structured response format, we forced JSON output containing: the assessed story, model details, and per-criterion quality assessments. Each assessment comprised a 1–4 quality score, an explanation, and a list of identified problems (with explanation, severity, and suggestions). We considered three tasks:

- INVEST: assessment constrained to the INVEST criteria (implicitly inferred by the LLM); focus on negative feedback (problems).
- RTI: an overall "ready to implement" judgment, based on the INVEST results.
- Custom DoR: assessment against company-specific Definition-of-Ready criteria (available for Company-1).

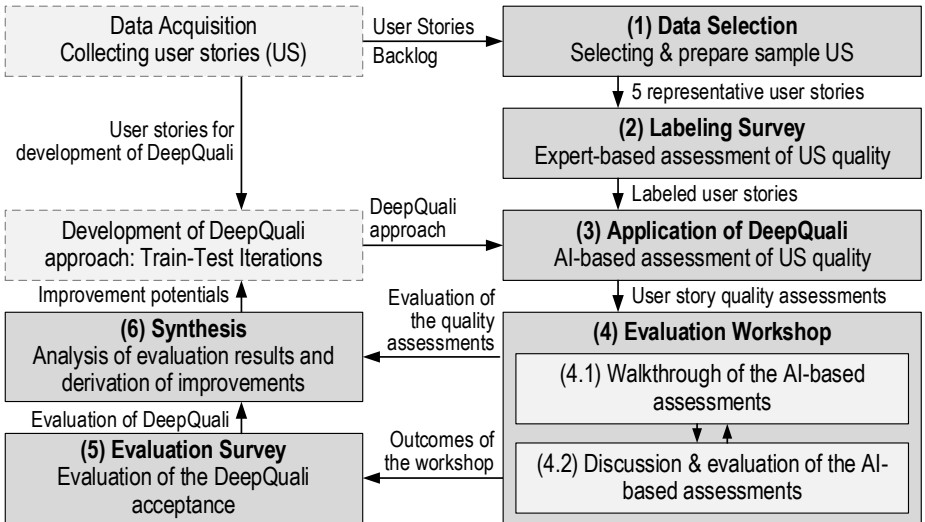

**Fig. 2.** Evaluation procedure.

**(4) Evaluation workshops**: The researcher demonstrated DeepQuali to the experts and walked them through the assessments for each sample story.

- Group discussion: Experts asked clarification questions and discussed strengths and weaknesses, focusing on (1) clarity and completeness of explanations and problems, (2) alignment with INVEST/DoR definitions, (3) consistency between explanations and numeric scores, and (4) consistency with their own expectations.
- Individual feedback survey: Experts evaluated DeepQuali's outputs (INVEST and RTI scores and explanations, and problem descriptions and criticality). Experts in Company-1 additionally rated DeepQuali's assessments against their custom DoR criteria (e.g., template use, level of detail of user story definition).

**(5) Evaluation survey**: We assessed user acceptance using an adapted UTAUT-2 framework [24], removing constructs inapplicable to this novel, immature tool (social influence, hedonic motivation, price value, experience/habit).

**(6) Synthesis**: We analyzed evaluation outcomes by transforming agreement-scale data to numerical scales, aggregating ratings using medians, and analyzing consistency

between experts using Spearman's Rho and Kendall's Tau. For RQ1, we computed absolute deviations and classification metrics (accuracy, precision, recall, F1-score) comparing DeepQuali's ratings against expert ground truth. For RQ2, we quantitatively analyzed experts' evaluations and qualitatively analyzed workshop feedback. For RQ3, we analyzed acceptance ratings using descriptive statistics and box plots. Due to small sample sizes, we did not calculate statistical significance tests.

Finally, we derived improvement opportunities and future research directions for applying GAI/LLMs to requirements quality assessment. Due to the small sample (three to five user stories, four experts), we did not perform statistical significance tests.

# 5 Results and Discussion

Due to time constraints, only three of five sample user stories were evaluated during the Company-1 workshop, which may influence perceived usefulness (RQ2) and acceptance (RQ3) compared to Company-2 where all five stories were evaluated.

**RQ1: Consistency between DeepQuali and expert assessments.**

*Agreement between experts:* Table 6 shows rather weak agreement between experts within each company. Experts agreed most on the *Small* and RTI criteria and disagreed most (in Company-1) on *Negotiability* and *Testability*. Different expertise areas and seniority levels may explain these inconsistencies.

**Table 2.** Agreement of users' assessments (bold: significant at p = 0.05).

|  |  | Independent | Nogotiable | Valuable | Estimable | Small | Testable | RTI | All |
|---|---|---|---|---|---|---|---|---|---|
| **Company-1** | *Tau* | 0,47 | 0,19 | 0,44 | 0,41 | **0,49** | 0,31 | 0,39 | **0,36** |
|  | *Rho* | 0,50 | 0,20 | 0,49 | 0,41 | **0,55** | 0,35 | 0,42 | **0,40** |
| **Company-2** | *Tau* | 0,31 | - | - | 0,34 | **0,54** | - | **0,54** | **0,46** |
|  | *Rho* | 0,32 | - | - | 0,35 | **0,55** | - | **0,61** | **0,50** |

*Consistency between experts and DeepQuali:* Fig. 3 illustrates deviations between assessments. For Company-1, lowest deviations occurred on *Estimable*; highest on *Negotiable* and *Testable*, where experts rated lower than DeepQuali.

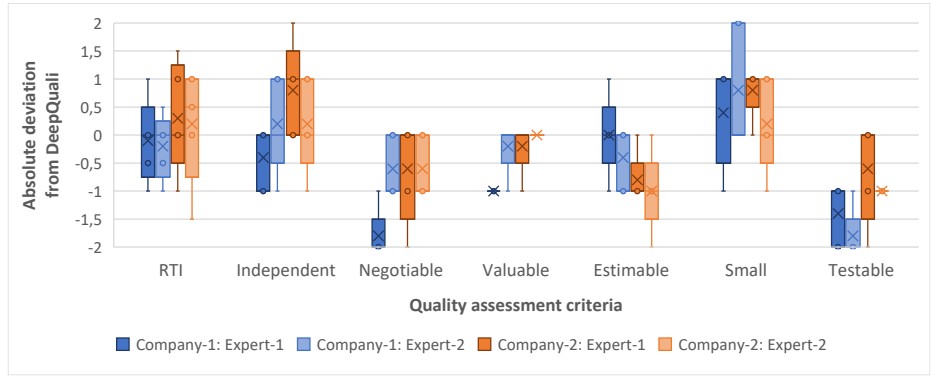

**Fig. 3.** Deviations between expert and DeepQuali quality assessments

More experienced experts tended to be more critical. For Company-2, lowest deviations were on *Valuable*; highest on *Estimable* and *Testable*. On the *Small* criterion, all experts rated higher than DeepQuali – possibly because detailed acceptance criteria or task lists in user stories led DeepQuali to suggest splitting them. Future improvements should incorporate company-specific standards for user story structure. Classification metrics confirmed these patterns.

**RQ2: Expert evaluation of DeepQuali assessments.**

Table 3 shows experts' evaluations. Company-1 experts were most critical regarding *actionability*, *completeness*, and *consistency* of INVEST and RTI assessments, but positively evaluated problem reporting and explanations. Company-2 experts gave higher evaluations overall. In both companies, more experienced experts were more critical. Experts appreciated explanations accompanying numerical ratings and RTI summaries for decision-making. However, they criticized ambiguous formulations ("it may," "it could"), perceived bias toward finding problems even when none existed, and suggested binary RTI assessments and evaluation of positive aspects alongside problems. They noted that assessing the *Independent* criterion requires considering the broader backlog context.

**Table 3.** Median evaluations of the DeepQuali assessments' quality

| | Company-1 (three user stories) | | | | | | | | Company-2 (five user stories) | | | | | |
| | INVEST | | RTI | | Problems | | DOR | | INVEST | | RTI | | Problems | |
| | E1 | E2 | E1 | E2 | E1 | E2 | E1 | E2 | E1 | E2 | E1 | E2 | E1 | E2 |
|---|---|---|---|---|---|---|---|---|---|---|---|---|---|---|
| Accurate | 3 | 2 | 3 | 2 | 3 | 3 | 3 | 2 | 4 | 3 | 4 | 4 | 3 | 3 |
| Actionable | 2 | 2 | 3 | 2 | 3 | 3 | 3 | 2 | 4 | 3 | 4 | 4 | 4 | 2 |
| Complete | 2 | 2 | 2 | 2 | 3 | 2 | 3 | 2 | 4 | 3 | 4 | 4 | 3 | 2 |
| Consistent | 2 | 2 | 3 | 2 | 3 | 3 | 3 | 2 | 3 | 3 | 3 | 3 | 3 | 3 |
| Context-driven | 3 | 3 | 3 | 3 | 3 | 2 | 3 | 1 | 4 | 4 | 4 | 4 | 4 | 4 |
| Explainable and Clear | 2 | 3 | 3 | 3 | 4 | 3 | 3 | 2 | 4 | 3 | 4 | 3 | 4 | 3 |
| Quality comform | 2 | 3 | 3 | 2 | 3 | 3 | 3 | 2 | 4 | 3 | 4 | 4 | 4 | 3 |
| Relevant | 3 | 2 | 3 | 3 | 3 | 4 | 3 | 2 | 4 | 4 | 4 | 4 | 4 | 4 |

**RQ3: Acceptance of DeepQuali.**

Fig. 4 shows acceptance survey results. Experts agreed on DeepQuali's potential usefulness, ease of use, and ability to improve work performance, expressing willingness to use it. However, they criticized insufficient facilitating conditions—an integrated tool appears critical for acceptance.

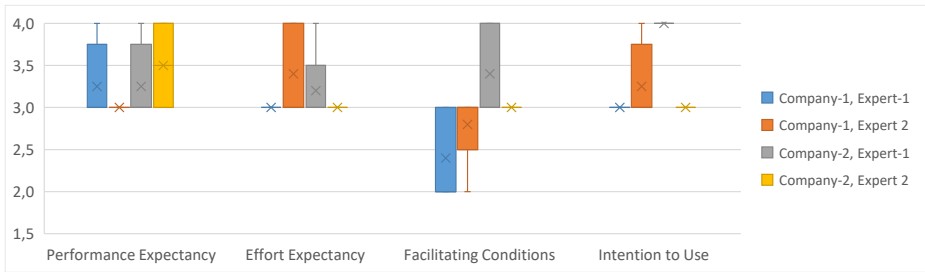

**Fig. 4.** Results of the survey on perceived user acceptance of DeepQuali.

## 6 Threats to Validity

*Construct validity* is threatened by the immaturity of DeepQuali and the early study stage: users were guided through the tool instead of using it independently, which limits the validity of the "user acceptance" construct. *Internal validity* threats arise from limited control over contextual factors, especially experts' background and experience, which clearly influenced their ratings and views on usefulness; future studies should explicitly account for such confounders. *Conclusion validity* is limited by the small number of experts and user stories, which prevents disentangling influences such as divergent interpretations or weaknesses of the INVEST criteria. Replicability is further challenged by the probabilistic nature of the LLM, although we reduced this risk via parameter tuning. *External validity* is restricted by the focus on two companies and two ASD projects; broader studies in different organizational and project contexts are needed to generalize the findings.

## 7 Summary and Conclusions

With DeepQuali, an LLM-based approach for assessing user story quality in ASD, we address a gap in using LLMs for requirements QA. We evaluated DeepQuali in two small software companies, each providing user stories and two subject matter experts. DeepQuali assessed quality using INVEST and custom Definition-of-Ready criteria, providing quantitative ratings, explanations, issue descriptions, and overall readiness assessments. The evaluation analyzed consistency with expert judgments and experts' views on usefulness and acceptance. While results are not statistically significant, they are promising. Experts valued the explanations, explicit quality criteria, and summary ratings for decision-making. However, they noted that DeepQuali sometimes identified non-existent problems and suggested it should also highlight positive aspects and consider project context. Experts agreed on DeepQuali's potential to improve their work and expressed willingness to use it if integrated into their work environments. For practitioners and researchers, our study offers initial evidence that LLMs can support requirements QA. Future work will improve DeepQuali based on feedback, extend capabilities to assess multiple user stories and additional quality aspects (completeness, consistency, redundancy, contradictions, coherence with epics), and address GAI uncertainties.

## Acknowledgments

Parts of this work were funded by the Federal Ministry of Research, Technology and Space (BMFTR) under grant no. 01IS23016D. We are very grateful to our participants.

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
