# OpenReview forum: "DeepQuali: Initial results of a study on the use of large language models for assessing the quality of user stories."
_KI/2026/Workshop/AI4SE — AI4SE Workshop_

### Official Review · Reviewer_2RsP · 2026-06-09
**An excellent presentation of an interesting topic. A valuable contribution to the planned workshop.**

**Rating:** 8
**Confidence:** 5

**Review:**

Empirically grounded case study on LLM-based quality assurance of user stories, considering the INVEST principle and specifically defined evaluation criteria.

Reflection on the content:

- The “DeepQual” approach reflects a scientifically grounded methodology. However, its practical application would benefit from stronger integration into existing software development processes.
- The case study on LLM-based quality assurance of user stories (Chapters 4 and 5) is clearly structured and presented in a comprehensible manner.
- Potential statistical limitations of the approach are transparently and convincingly discussed in a dedicated chapter.

Potential suggestions for improvement:

- selection of the LLM versions used could be presented more transparently, particularly with regard to the criteria underlying their selection.
- Potential risks associated with LLM-supported quality assurance, such as security and privacy concerns, should be addressed more thoroughly.
- The paper could provide a more detailed discussion of domain-specific challenges related to the use of LLMs, as these often present significant issues in practice.

---

### Official Review · Reviewer_iBNZ · 2026-06-12
**Relevant and interesting paper on usage of LLMs for RE**

**Rating:** 7
**Confidence:** 4

**Review:**

Relevance of paper:
The paper investigates an LLM-based approach called DeepQuali for assessing and improving requirements quality in agile projects. The proposed approach is evaluated with 4 experts from two different companies. The focus of this research is to understand the performance of the LLM in comparison to human experts, as well as the usefulness and perceived acceptance of the LLM-based approach.
The topic is highly relevant for research and practice, as it addresses an important challenge in the early stages of SE by focusing on the quality of requirements (user stories).

Structure and clarity of paper:
Overall, the paper is written well and follows a logical structure – both makes this paper easy to read and understand.

Related work:
The paper addresses relevant related work in the domain. What would help to position it even more in its research context and practical relevance is adding some additional references wrt. to the current state of the practice and potential requirements engineering solutions that exist in the market to support RE and quality assurance of requirements. The paper states “Despite diverse tool support [23], RE remains complex and time-consuming” – this could be elaborated in more detail where the gaps exist, and it should be discussed how DeepQuali potentially supports / links into the existing state of practice.

Research methodology:
Overall the research methodology is well described and the procedure is easy to understand and follow. I have two concerns with regards to the measurement instrument: (1) Literature references are provided but the measurement items (selection) is not clearly specified. Please clarify which and how many items where used per question and if items where modified. (2) The authors use a 4-point likert scale. In user-focused studies of perceptions 5- or even 7-point likert scales are the standard. Is there a reason to deviate from this? This would require explanation.